# Characterization of Light-Enhanced Respiration in Cyanobacteria

**DOI:** 10.3390/ijms22010342

**Published:** 2020-12-31

**Authors:** Ginga Shimakawa, Ayaka Kohara, Chikahiro Miyake

**Affiliations:** 1Department of Biological and Environmental Science, Faculty of Agriculture, Graduate School of Agricultural Science, Kobe University, 1-1 Rokkodai, Nada, Kobe 657-8501, Japan; a.kohara1199@gmail.com (A.K.); cmiyake@hawk.kobe-u.ac.jp (C.M.); 2Core Research for Environmental Science and Technology, Japan Science and Technology Agency, 7 Goban, Chiyoda, Tokyo 102-0076, Japan

**Keywords:** oxygen, light-enhanced respiration, photosynthesis, respiratory terminal oxidases

## Abstract

In eukaryotic algae, respiratory O_2_ uptake is enhanced after illumination, which is called light-enhanced respiration (LER). It is likely stimulated by an increase in respiratory substrates produced during photosynthetic CO_2_ assimilation and function in keeping the metabolic and redox homeostasis in the light in eukaryotic cells, based on the interactions among the cytosol, chloroplasts, and mitochondria. Here, we first characterize LER in photosynthetic prokaryote cyanobacteria, in which respiration and photosynthesis share their metabolisms and electron transport chains in one cell. From the physiological analysis, the cyanobacterium *Synechocystis* sp. PCC 6803 performs LER, similar to eukaryotic algae, which shows a capacity comparable to the net photosynthetic O_2_ evolution rate. Although the respiratory and photosynthetic electron transports share the interchain, LER was uncoupled from photosynthetic electron transport. Mutant analyses demonstrated that LER is motivated by the substrates directly provided by photosynthetic CO_2_ assimilation, but not by glycogen. Further, the light-dependent activation of LER was observed even with exogenously added glucose, implying a regulatory mechanism for LER in addition to the substrate amounts. Finally, we discuss the physiological significance of the large capacity of LER in cyanobacteria and eukaryotic algae compared to those in plants that normally show less LER.

## 1. Introduction

Respiration is one of the most important biological activities, even in oxygenic phototrophs that produce NAD(P)H and ATP by photosynthetic metabolism. In plants, algae, and cyanobacteria, respiratory metabolism and electron transport systems are mostly conserved, except for some components [1,2]. In principle, respiration plays a role in producing ATP outside of chloroplasts, in the nonphotosynthetic tissues and under nonphotosynthetic conditions (i.e., darkness). However, it has been broadly suggested that respiration also interacts with photosynthesis during illumination of redox, metabolic, and energetic events in photosynthetic organisms [3,4,5,6,7,8,9]. The phenotypes of various respiratory mutants indicate that the interplay between respiration and photosynthesis is essential for the growth of photosynthetic organisms [10,11,12].

Physiological functions of respiration in association with photosynthesis have not yet been fully understood in photosynthetic organisms. In a variety of eukaryotic algae, the O_2_ uptake rate during dark respiration is frequently higher after illumination and can then reach about half of the photosynthetic O_2_ evolution rate. The light-enhanced dark respiration (here we termed it as light-enhanced respiration, LER), one of the most indisputable phenomena that shows respiration is linked to photosynthesis under light conditions, is associated with the amounts of respiratory substrates generated during photosynthesis in eukaryotic algae [4,13,14,15,16]. However, the details of the initiation and relaxation mechanism of LER are still unclear.

The interaction between respiration and photosynthesis has already been established in cyanobacteria, the prokaryotic algae broadly recognized as the progenitor of oxygenic photosynthesis [17,18,19]. Due to the lack of organelles, both respiration and photosynthesis proceed in one cell, and they both depend on the cytosolic metabolism and on the electron transport chain in the thylakoid membrane (Figure 1). The redox states of the photosynthetic electron transport components and the amounts of photosynthetic metabolites are affected in respiratory mutants [20,21,22,23,24,25,26], the interplay between respiration and photosynthesis in cyanobacterial cells. In the cyanobacterium *Synechocystis* sp. PCC 6803 (S. 6803), dark respiration starts from the degradation of glycogen catalyzed by debranching enzyme- or phosphorylase-dependent pathways [27,28]. Hexoses such as glucose are metabolized to pyruvate in the cytosol via glycolysis, similar to photosynthetic eukaryotes, and then drive the cyanobacterial tricarboxylic acid (TCA) cycle in the cytosol. Unique metabolic pathways in the cyanobacterial TCA cycle are still being uncovered, but the TCA cycle is likely to be the dominant source of NAD(P)H for the respiratory electron transport chain [29,30]. Meanwhile, in cyanobacteria oxidative the pentose phosphate pathway is also suggested to be an important source for reducing power, especially NADPH [31]. The thylakoid membrane contains both the photosynthetic and respiratory electron transport chains, both of which contribute to ATP production catalyzed by ATP synthase [32,33]. In the photosynthetic electron transport chain, the electrons generated by H_2_O oxidation at photosystem II (PSII) are transported to NADP^+^ at the electron acceptor side of PSI. In the respiratory electron transport, the electrons are transported from NAD(P)H dehydrogenase (NDH)-1, NDH-2, and succinate dehydrogenase (SDH) to cytochrome *c* oxidase (COX), reducing O_2_ to H_2_O. These two electron transport chains share the interchain components plastoquinone (PQ), the cytochrome *b*_6_/*f* complex, plastocyanin, and cytochrome *c* [18,22]. Additionally, in S. 6803, the cytochrome *bd*-type quinol oxidase complex (Cyd) reduces O_2_ to H_2_O using the electrons from plastoquinol [34]. Furthermore, akin to Cyd, the alternative respiratory terminal oxidase (ARTO) on the plasma membrane is reported to reduce O_2_ [35]. In addition to ARTO, some respiratory electron transport components also function on the plasma membrane, separate from photosynthetic electron transport, details of which remain debatable [22,33]. Despite the prokaryotic characteristics, the physiological functions of respiration under light conditions are still unclear, and the notable phenomenon, LER, has never been targeted in cyanobacteria. Here, we first characterize LER in S. 6803 to deepen the understanding on the molecular mechanism of the interactions between respiration and photosynthesis in photosynthetic organisms.

## 2. Results

### 2.1. Relationship of Light-Enhanced Respiration with Photosynthesis in S. 6803

Since LER is activated by illumination, it is difficult to analyze the capacity of LER in photosynthetic organisms that are already adapted to light. To maximize the observed capacity of LER, in this study, cyanobacterial cells were adapted to darkness for 24 h in advance to physiological measurements (see Section 4). After the long dark adaptation, almost all of the respiratory substrates were assumed to be consumed, and the dark respiration rate was approximately 5 µmol O_2_ mg chlorophyll (Chl)^−1^ h^−1^ (Figure 2), which was lower than the values previously reported for the wild type S. 6803 [21]. Upon illumination, net photosynthetic O_2_ evolution was observed, but the rate gradually decreased from 80 to 50 µmol O_2_ mg Chl^−1^ h^−1^ during illumination (Figure 2). After the actinic light was switched off, the O_2_ uptake rate was approximately 40 µmol O_2_ mg Chl^−1^ h^−1^, which was designated as LER in this study. The LER was relieved within 20 min, and finally, the O_2_ uptake rate was close to 10 µmol O_2_ mg Chl^−1^ h^−1^ (Figure 2 and Appendix A), which was consistent with the values previously detected in the S. 6803 wild type [21].

Next, we analyzed LER after different illumination periods. Contrary to the decrease in net photosynthetic O_2_ evolution rate, a increase in LER was observed with the time of illumination (Figure 3A). If any O_2_-dependent alternative electron transports do not compete with the CO_2_ assimilation in this situation, the sum of net O_2_ evolution and LER rates can be defined as the total photosynthetic O_2_ evolution rate corresponding to the gross photosynthetic activity. The total photosynthetic O_2_ evolution rate was kept almost constant and correlated with the effective quantum yield of PSII, Y(II), during illumination (Figure 3B). Overall, the decrease in the net O_2_ evolution rate reflected the increase in LER but not an inactivation of CO_2_ assimilation.

Further, the relationship of LER with the total photosynthetic O_2_ evolution rate was evaluated after a 10 min illumination with the actinic light at various light intensities in S. 6803 (Figure 4). The proportional relationship between LER and the total photosynthetic O_2_ evolution rate indicates that LER is activated, paralleled with the photosynthetic CO_2_ assimilation. In this study, LER was also observed in the other cyanobacterial strain *Synechococcus elongatus* PCC 7942 (S. 7942), also showing the proportional relationship between LER and the total photosynthetic O_2_ evolution rate (Figure 4). As a result, LER is likely to be a common phenomenon in cyanobacteria. However, the capacity of LER, based on the photosynthetic activity, was larger in S. 7942 than in S. 6803 (Figure 4).

### 2.2. Molecular Mechanism of Light-Enhanced Respiration in S. 6803

The correlation between LER and illumination time suggests that LER was motivated by reducing cofactors or metabolites generated during photosynthesis in the cyanobacterium S. 6803. We constructed the S. 6803 mutants deficient in respiratory components encoded in *cox* (COX), *cyd* (Cyd), *arto* (ARTO), *glgP* (glycogen phosphorylases), and *glgX* (glycogen deblanching enzymes) to investigate the molecular mechanism of LER in S. 6803 (Appendix A). In Figure 5A, we show the LER activity upon switching the actinic light off after 20 min of illumination in the wild type of S. 6803, the triple mutant for respiratory terminal oxidases (Δ*cox/cyd/arto*), and the quadruple mutant for glycogen phosphorylases and blanching enzymes (Δ*glgP1/glgP2/glgX1/glgX2*, termed Δ*glgP/glgX*). These respiratory mutants show a trend of lower dark respiration than the wild type before the illumination (Figure 5A). The capacity of LER was significantly lower in Δ*cox/cyd/arto*, but not different in Δ*glgP/glgX* compared to the wild type (Figure 5). In Δ*cox/cyd/arto*, approximately 60% of the capacity was lost, but a part of the LER was still detected even in this mutant (Figure 5A). Nevertheless, there was only a minimal change in the net O_2_ evolution rate in Δ*cox/cyd/arto* after the actinic light was turned off, suggesting that the transient increase in O_2_ uptake was almost derived from these respiratory terminal oxidases (Figure 5B and Appendix A). It has been suggested that a leak of electrons to O_2_ in the dark produces reactive oxygen species in a light–dark cycle in the S. 6803 mutants deficient in respiratory terminal oxidases [22], which may partially contribute to the O_2_ uptake observed in Δ*cox/cyd/arto* (Figure 5A).

We also analyzed LER in the S. 6803 mutants deficient in components related to O_2_-dependent alternative electron transport [36,37]. Flavodiiron proteins (FLVs) function as alternative electron sinks at the electron acceptor side of PSI to dissipate the excess electrons from reducing ferredoxin to O_2_ in S. 6803 [38,39]. Disruption of FLV had no effect on LER (Figure 6A). Further, we also assessed the possibility that photorespiration partially contributes to LER. In all plants, except the C_4_ plants, postillumination transient O_2_ uptake is largely observed at a CO_2_ compensation point and is driven by photorespiration [40,41]. In S. 6803, similar to eukaryotic algae [14], the capacity of LER was rather low under CO_2_ limitation, and there was no difference between the wild type and the photorespiratory mutant deficient in glycolate dehydrogenase (Figure 6B). Overall, these mutant analyses suggested that O_2_-dependent alternative electron transport such as FLV-mediated Mehler-like reactions and photorespiration are not included in the LER.

To investigate the effects of the respiratory substrates on LER in S. 6803, we exogenously added glucose (final concentration, 5 mM) 10 min before illumination. The cyanobacterium S. 6803 can take up exogenous glucose as a respiratory substrate to accumulate a variety of intermediates in the glycolysis and the Calvin–Benson cycle [21]. In fact, the dark respiration rate increased to approximately 20 µmol O_2_ mg Chl^−1^ h^−1^ in the presence of glucose (Figure 7). We confirmed that the dark respiration rate did not further increase with 15 mM glucose, indicating that 5 mM glucose causes enough saturation to enhance the dark respiration in S. 6803. The amount of glucose consumed by LER is roughly estimated from the O_2_ consumption—for example, in Figure 2, it can be seen that only 8 μM was produced by 10 min in dark. We note that the exogenous addition of glucose does not necessarily mimic the situation where the whole pool of respiratory substrates is saturated. Although the capacity of LER was greater with exogenously added glucose, light-dependent activation was observed, as was the case without glucose (Figure 3A and Figure 7B). Overall, the increase in LER paralleled with photosynthesis can not only be explained by the amounts of respiratory substrates. The effect of exogenous glucose on LER was also investigated at various light intensities of the actinic light. Even in the presence of glucose, the capacity of LER increased with the light intensity as in the manner without glucose (Appendix A), which also suggested that there is a certain impact of light on increasing the LER capacity regardless of the amount of glucose.

We further tested the relaxation of the effect of exogenous glucose on LER in the darkness. Although net O_2_ uptake rate reached approximately 50 µmol O_2_ mg Chl^−1^ h^−1^ after the 10 min illumination with actinic light in the presence of glucose (Figure 7B), it decreased to 20 µmol O_2_ mg Chl^−1^ h^−1^ if glucose was added 70 min after the actinic light was turned off (Figure 8A), which was almost the same value as that from when glucose was added before the illumination (Figure 7). Here, we hypothesize an elusive light-dependent activation step of LER and assessed the relaxation by adding glucose at different times in dark after the 10 min illumination (Figure 8B), indicating that the stimulation of O_2_ uptake by exogenous glucose was enhanced by the illumination and slowly relaxed in dark to a constant level in approximately 1 h.

## 3. Discussion

### 3.1. Cyanobacterial Light-Enhanced Respiration Is Metabolically Coupled with Photosynthesis

Simultaneously with the evolution of PSII, O_2_ is consumed under light through respiration, photorespiration, and an FLV-mediated Mehler-like reaction [42]. The large amount of O_2_ consumed after an actinic light is turned off corresponds to the amount of substrate for respiration or photorespiration produced in the light [41]. Therefore, the analysis of postillumination transient O_2_ uptake is a useful method to uncover the molecular mechanism of O_2_-consuming reactions during photosynthesis. Whereas in plant leaves the postillumination transient O_2_ uptake is dominantly driven by photorespiration under CO_2_ limitation [40,41], in eukaryotic algae it is mainly derived from LER where enough CO_2_ is available [14,16]. Here, we analyzed the postillumination transient O_2_ uptake in cyanobacteria and found that it is derived from LER, similar to eukaryotic algae, and is associated with photosynthesis (Figure 3 and Figure 4). In cyanobacteria, respiration and photosynthesis both depend on their metabolism, electron transport, and redox signaling in one cell. Due to the sharing of the interchain components, the redox states of PQ, PSI, and NADP^+^ are potentially affected by defects in the respiratory electron transport components in the darkness and light–dark transition [20,43,44]. In this study, we observed O_2_ uptake rate of LER that can be comparable to the net photosynthetic O_2_ evolution rate (Figure 2). The increase in LER caused the decrease in net O_2_ evolution rate (Figure 3A). The sum of net O_2_ evolution and LER was coupled with Y(II) (Figure 3B). Overall, the large capacity of LER originated from the respiratory electron transport chain driven by substrates produced in the Calvin–Benson cycle, but not by an electron leakage from photosynthetic electron transport chain on the thylakoid membrane. This conclusion was consistent with the results of mutant analyses (Figure 5 and Figure 6) and is in agreement with the fact that the electron sink capacity of the alternative electron transport from PSII to respiratory terminal oxidases is not significant in light, compared with photosynthetic linear electron flow and FLV-mediated electron transports [36,45]. Based on the present dataset, we cannot exclude the possibility that the LER measured after the actinic light was turned off did not necessarily reflect the exact respiratory O_2_ uptake during illumination since the membrane potential, also called *proton motive force*, depends on both the respiratory and photosynthetic electron transport chains in the thylakoid membrane [46], which possibly suppresses the respiratory electron transport via “back-pressure” of the *proton motive force* [47]. Meanwhile, the net O_2_ evolution rate decreased with Y(II) being kept constant (Figure 2 and Figure 3), supporting the hypothesis that LER exactly occurs during the illumination.

### 3.2. Cyanobacterial Light-Enhanced Respiration Is Not Restricted Only to Respiratory Substrate Amounts

Characterization of LER in S. 6803 led us to analyze the respiratory mutants Δ*cox/cyd/arto* and Δ*glgP/glgX*. The same extent of LER in Δ*glgP/glgX* as in the wild type clearly indicated that LER requires substrates derived not from glycogen, but directly from the Calvin–Benson cycle. There are several possibilities for electron donation from the stroma to the respiratory electron transport chain. The amounts of specific substrates can be an important factor for enhancing and suppressing respiration. For example, it has recently been reported that the NADPH-generating enzymes in the oxidative pentose phosphate pathway are inhibited by the TCA cycle intermediate citrate in S. 6803, implying a regulation for balancing NAD(P)H production in cyanobacterial respiratory metabolism [48]. It may be reasonable to assume that the activities of some respiratory enzymes are strictly regulated by specific metabolites to control the complex mixture of various respiratory and photosynthetic metabolisms in one cyanobacterial cell. The addition of exogenous glucose increased the dark respiration rate, but it did not affect the light-dependent activation process of LER (Figure 7B and Appendix A), implying that the regulatory mechanism of cyanobacterial LER does not only depend on the amounts of respiratory substrates. Based on the experiments with exogenous glucose, the elusive light-dependent activation step almost finished within 20 min in light and was then slowly relaxed to a constant level within 1 h in dark. The redox state of the components related to photosynthetic electron transport (e.g., PQ) may function in regulating the capacity of LER through a signaling system. Meanwhile, we also note the possibility that LER is stimulated by a specific metabolite produced by photosynthesis in light, independently of the metabolism driven by glucose, in the same activation manner with and without exogenous glucose (Figure 7B and Appendix A).

### 3.3. Light-Enhanced Respiration Plays a Role in Producing ATP through Consumption of Photosynthates

It is assumed that the physiological significance of LER is the immediate production of ATP using excess photosynthates. In other words, ATP is additionally demanded even during photosynthesis, probably for a variety of cell metabolisms. The present study showed that the LER capacity in S. 6803 can reach approximately half of the total photosynthetic O_2_ evolution rate, which is likely to be sufficient to consume excess organic acids produced by photosynthesis under light conditions. However, the cyanobacterium S. 7942 mutant deficient in glycogen synthesis excretes organic acids such as pyruvate and 2-oxoglutarate in the medium, thereby increasing the concentration of exogenous pyruvate at approximately 0.1 mM per day under nitrogen deprivation [49]. This implies that even at a greater capacity of LER, cyanobacteria assimilate CO_2_ to sugars at a rate that is more than what is needed and store them in the form of glycogen during photosynthesis. Possibly, the higher photosynthetic productivity as opposed to the demand for cell growth could have led to the evolution of cyanobacteria to multicellular photosynthetic organisms.

### 3.4. Evolutionary Changes of Light-Enhanced Respiration and Photorespiration Capacities from Algae to Plants

In the present study, cyanobacteria possess a large capacity of LER, similar to eukaryotic algae, including green algae and a variety of secondary algae. That is, LER is a common phenomenon in prokaryotic and eukaryotic algae, even though there are variations in the capacities among species. Meanwhile, the capacity of LER is much lower in in vivo plant leaves. The difference between these unicellular algae and plant leaves could be due to the different sink capacities for photosynthates. In cyanobacteria and eukaryotic algae, the utilization of photosynthates can often be limited to carbon metabolism in cells, especially under higher light and CO_2_ conditions. The productivity of photosynthetic CO_2_ assimilation has been developed during the evolution of photosynthetic organisms to shallow and terrestrial fields, resulting in a higher accumulation of photosynthates. In particular, out of water, photosynthetic organisms can have difficulties in excreting excess photosynthates. These factors could explain why photosynthetic organisms evolved to multicellular formations and developed intercellular mass transfers to expand the carbon sink capacity. Contrary to the evolution of terrestrial plants in the photosynthetic green lineage, some cyanobacteria, such as *Prochlorococcus*, are likely to have adopted the strategy to develop carbon excretion for the acclimation to oceanic surface ecosystems [50].

In contrast to LER, photorespiration has increased the energetic and metabolic capacity reflected in the O_2_ uptake in the evolutionary history from algae to plants [40]. In land plants, except for C_4_ plants, the large capacity of postillumination transient O_2_ uptake is observed under CO_2_ limitation, which is different from cyanobacteria (Figure 6B) and eukaryotic algae [14]. Indeed, the O_2_ uptake in plants mainly originates from photorespiration [41]. Although photorespiratory metabolism is almost conserved and plays an important role in the growth under CO_2_ limitation in S. 6803 [51], the electron flux capacity through photorespiration is negligible, as compared to the photosynthetic CO_2_ assimilation [36], which was also demonstrated by the measurement of LER in the present study (Figure 6B).

## 4. Materials and Methods

### 4.1. Growth Conditions and Chl a Determination

The cyanobacterium S. 6803 was cultured as described in our previous work [52]. Cells from the culture, with initial OD_750_ of 0.1 to 0.2, were inoculated into liquid BG-11 medium and grown on a rotary shaker (100 rpm) under continuous fluorescent lighting (25 °C, 150 μmol photons m^−2^ s^−1^) at 2% (*v*/*v*) [CO_2_]. The optical density of the medium was measured using a spectrophotometer (U-2800A, Hitachi, Tokyo, Japan). For all the physiological measurements of LER, cells from the exponential growth phase were adapted to the dark for 24 h before they were used.

For Chl measurement, cells from the 0.1 to 1.0 mL cultures were centrifugally harvested and resuspended by vortexing in 1 mL 100% (*v*/*v*) methanol. After incubation for 5 min in the dark, the suspension was centrifuged at 10,000× *g* for 5 min. The total Chl *a* was spectrophotometrically determined from the supernatant [53].

### 4.2. Generation of Mutants

The mutants of S. 6803, Δ*cox/cyd*, Δ*glgP*, Δ*flv*, and Δ*glcD* were constructed in our previous studies [21,36,37,43]. To disrupt the genes *ctaCII* (*arto*; *sll0813*), *glgX1* (*slr0237*), and *glgX2* (*slr1857*) in S. 6803, each portion of the coding region was replaced with streptomycin and erythromycin resistance cassettes (*Sm*^r^ and *Em*^r^), derived from pRL453 and pRL425, respectively [54]. The triple mutant for the respiratory terminal oxidases was generated by the transformation of Δ*cox/cyd* with the *arto*-deleted construct containing *Em*^r^. The quadruple mutant Δ*glgP/glgX* was generated by the two transformations of Δ*glgP* with the *glgX1-* and *glgX2*-deleted constructs containing *Sm*^r^ and *Em*^r^. For the preparation of the construct with *Sm*^r^, the coding region was cloned into a pTA2 vector (Toyobo, Otsu, Japan). The plasmid was linearized by inverse polymerase chain reaction (PCR) and ligated with *Sm*^r^ [55]. To prepare the constructs with *Em*^r^, two separated PCR fragments for the coding regions were linked with *Em*^r^ by successive PCR [36,56]. Transformants were selected on 0.5% (*w*/*v*) agar plates of BG-11 medium containing antibiotics. Complete segregation of each mutant was confirmed by PCR (Appendix A). The primers used for the amplification of each region are shown in Appendix A.

### 4.3. Measurement of O_2_ and Chl Fluorescence

Net uptake and evolution of O_2_ were measured concurrently with Chl fluorescence using a Clark-type O_2_ electrode (Hansatech Instruments Ltd., King’s Lynn, UK). Cell samples in the reaction mixture (2 mL, 50 mM HEPES-KOH, pH 7.5, 10 mM NaHCO_3_, 10 μg Chl mL^−1^) were stirred with a magnetic microstirrer and illuminated with red actinic light (*λ* >620 nm) at 25 °C. The photon flux densities are indicated in the figure legends. A halogen lamp (Xenophot HLX 64625, Osram, München, Germany) from an LS2 light source (Hansatech) was used as the red actinic light source. The rates of O_2_ uptake and evolution were calculated from the change in the O_2_ concentration in the mixture in 10 s (examples of raw traces are shown in Appendix A). The data points were acquired every second.

The relative Chl fluorescence originating from Chl *a* was measured using a PAM-Chl fluorometer (PAM-101; Walz, Effeltrich, Germany) as previously described [57]. Pulse-modulated excitation was achieved using an LED lamp with a peak emission at 650 nm. Modulated fluorescence was measured at *λ* > 710 nm (Schott RG9 long-pass filter; Walz). The steady-state fluorescence (F_s_) was monitored under actinic light. To determine the maximum variable fluorescence in the light (F_m_′), 1000 ms pulses of saturated light (10,000 μmol photons m^−2^ s^−1^) were supplied. The fluorescence terminology used in this study follows the previous report [58]. Y(II), the effective quantum yield of PSII, was defined as (F_m_′ − F_s_)/F_m_′.

## Figures and Tables

**Figure 1 ijms-22-00342-f001:**
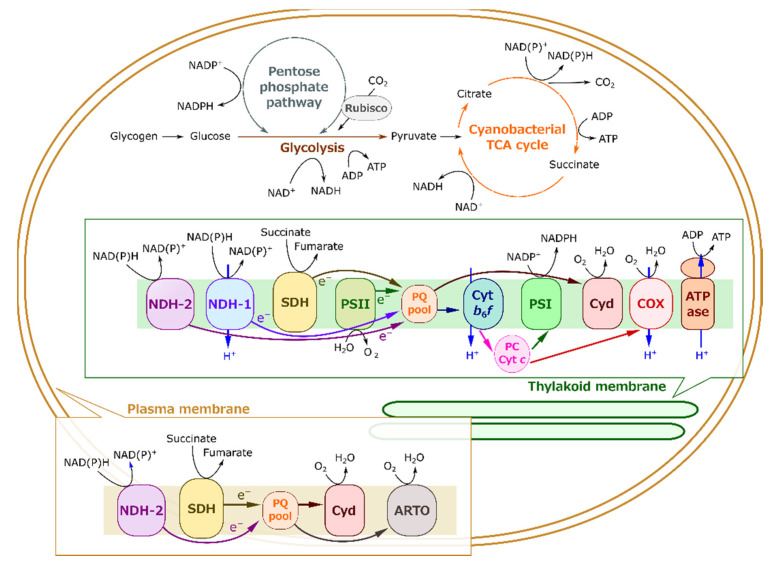
A brief illustration of respiration and photosynthesis sharing their metabolisms and electron transport chains in one cyanobacterial cell. To simplify the illustration, stoichiometry of each reaction and many bypasses are disregarded here. The direction of the arrows mainly follows the respiration mode. Under light conditions, photosynthetic electron transport proceeds from PSII to PSI, and Rubisco assimilates CO_2_ to initiate the reductive pentose phosphate pathway (also called the Calvin–Benson cycle). Conversely, the oxidative pentose phosphate pathway proceeds in sugar catabolism, producing reducing power as NADPH. Abbreviations for enzymes and the complexes are as follows: Rubisco, ribulose 1,5-bisphosphate carboxylase/oxygenase; NDH-1 and -2, type I and II NAD(P)H dehydrogenase; SDH, succinate dehydrogenase; PSII and PSI, photosystem II and I; PQ, plastoquinone; Cyt *b*_6_*f*, cytochrome *b*_6_*f* complex; PC, plastocyanin; Cyt *c*, cytochrome *c*; Cyd, cytochrome *bd*-type quinol oxidase; COX, cytochrome *c* oxidase; ATPase, ATP synthase; ARTO, alternative respiratory terminal oxidase.

**Figure 2 ijms-22-00342-f002:**
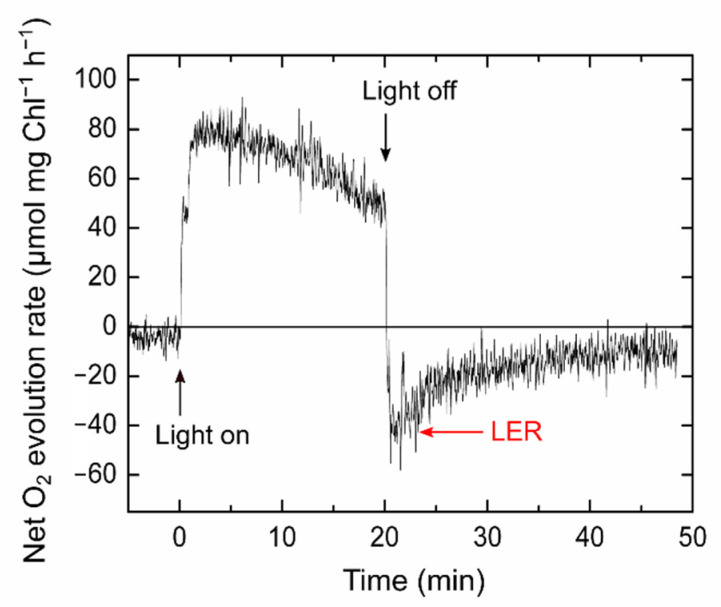
Light-enhanced respiration (LER) in *Synechocystis* sp. PCC 6803. The cyanobacterial cells adapted to darkness for 24 h (10 µg Chl mL^−1^) were illuminated with a red actinic light (190 μmol photons m^−2^ s^−1^) for 20 min as indicated by black arrows. We defined the O_2_ uptake rate just after the light was turned off as LER (red arrow). The representative data (*n* = 9, biological replicates) are shown.

**Figure 3 ijms-22-00342-f003:**
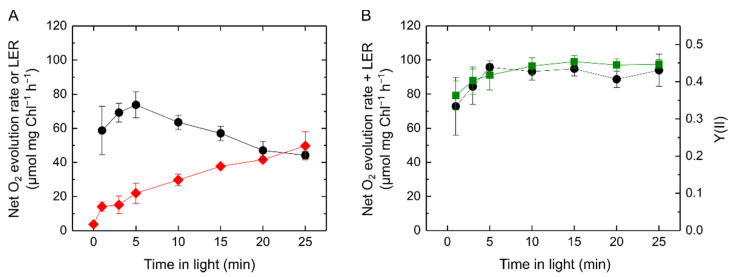
Light-enhanced respiration (LER) at different illumination times in *Synechocystis* sp. PCC 6803 adapted to the darkness for 24 h (10 µg Chl mL^−1^). (**A**) Net O_2_ evolution rate (black circles) and LER (red diamonds) by the illumination with a red actinic light (190 μmol photons m^−2^ s^−1^). (**B**) Comparison of the total photosynthetic O_2_ evolution rate (black circles) defined as the sum of net O_2_ evolution rate and LER with effective quantum yield of photosystem II, Y(II) (green squares). Data are shown as the mean with the standard deviation (*n* = 4, biological replicates).

**Figure 4 ijms-22-00342-f004:**
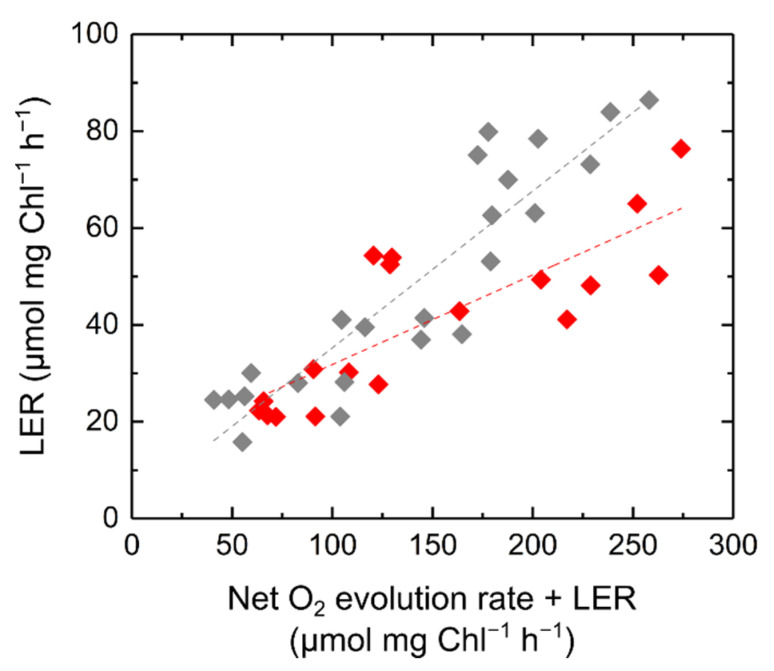
Relationship of light-enhanced respiration (LER) with the total photosynthetic O_2_ evolution rate defined as the sum of net O_2_ evolution rate and LER in *Synechocystis* sp. PCC 6803 (red) and *Synechococcus elongatus* PCC 7942 (grey) adapted to the darkness for 24 h. The cyanobacterial cells (10 µg Chl mL^−1^) were illuminated with a red actinic light for 10 min at 90 (*n* = 4), 190 (*n* = 7), 400 (*n* = 3), and 600 (*n* = 4) μmol photons m^−2^ s^−1^ (biological replicates). Dashed lines represent the estimated linear regressions of the data (S. 6803, *R*^2^ = 0.6586; S. 7942, *R*^2^ = 0.8305).

**Figure 5 ijms-22-00342-f005:**
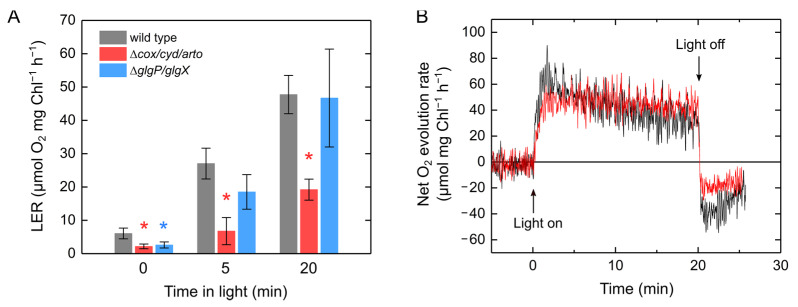
Light-enhanced respiration (LER) in wild type (grey bars) and the mutant of *Synechocystis* sp. PCC 6803 deficient in respiratory terminal oxidases (Δ*cox/cyd/arto*, red bars) and glycogen degradation enzymes (Δ*glgP/glgX*, blue bars) adapted to the darkness for 24 h. (**A**) LER after 20 min illumination with a red actinic light (190 μmol photons m^−2^ s^−1^). Data are shown as the mean with the standard deviation (*n* = 4, biological replicates). Asterisks indicate statistically significant differences (*p* < 0.05) between wild type and the mutants as per Student’s *t*-test. (**B**) Representative comparison of the net O_2_ evolution rate in the wild type (black) with that in the Δ*cox/cyd/arto* (red).

**Figure 6 ijms-22-00342-f006:**
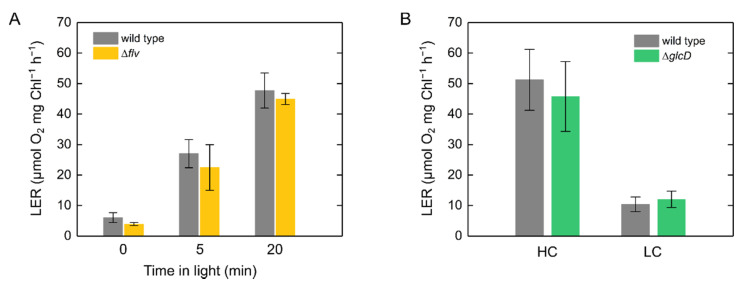
Light-enhanced respiration (LER) in wild type (grey bars) and the mutants of *Synechocystis* sp. PCC 6803 deficient in components related to alternative electron transport pathways. (**A**) LER in the mutant that lacks flavodiiron proteins 1, 3, and 4 (Δ*flv*, yellow bars). (**B**) LER in the mutant deficient in glycolate dehydrogenase 1 and 2 (Δ*glcD*, green bars). In “HC (high CO_2_)”, cyanobacterial cells adapted to the darkness for 24 h (10 µg Chl mL^−1^) were illuminated with red actinic light (190 μmol photons m^−2^ s^−1^) for 20 min, as in the LER measurements in the other strains. The “LC (low CO_2_)” indicates the condition where available CO_2_ is limited, which was prepared by a long-term (1–2 h) illumination without adding NaHCO_3_ to the oxygen electrode chamber (Shimakawa et al., 2015). The actinic light was turned off at the steady state under CO_2_ limitation, and then LER was measured. Data are shown as the mean with the standard deviation (*n* = 4, biological replicates).

**Figure 7 ijms-22-00342-f007:**
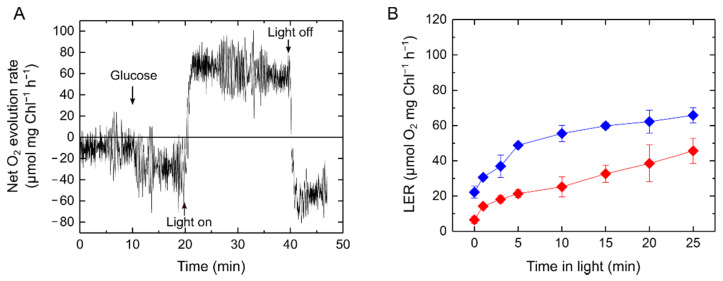
Effects of exogenously added glucose on light-enhanced respiration (LER) in *Synechocystis* sp. PCC 6803 adapted to the darkness for 24 h (**A**). Representative data of net O_2_ evolution rate in response to the illumination with a red actinic light (190 μmol photons m^−2^ s^−1^). Glucose (5 mM) was exogenously added 10 min before the start of the illumination as indicated by a black arrow (**B**). LER at different illumination times in the absence (red) and presence (blue) of glucose. Data are shown as the mean with the standard deviation (*n* = 3, biological replicates).

**Figure 8 ijms-22-00342-f008:**
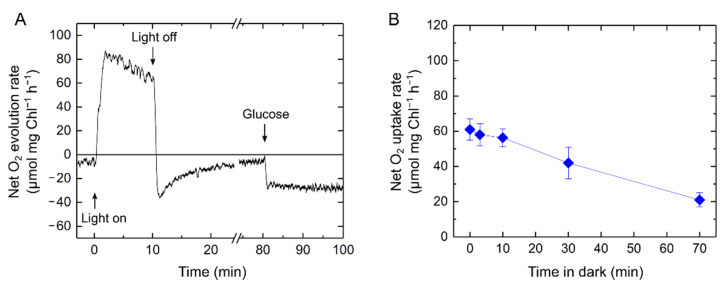
Relaxation of effects of exogenously added glucose on light-enhanced respiration (LER) in the darkness in *Synechocystis* sp. PCC 6803. The cyanobacterial cells adapted to darkness for 24 h (10 µg Chl mL^−1^) were illuminated with a red actinic light (190 μmol photons m^−2^ s^−1^) for 10 min. (**A**) Representative data of net O_2_ evolution rate in response to the illumination with the actinic light and the addition of glucose (5 mM) as indicated by arrows. (**B**) Net O_2_ uptake rate stimulated by the addition of glucose at different times after the actinic light was turned off. At time zero, glucose was exogenously added and then the O_2_ uptake was measured just after the 10 min illumination with the actinic light. Data are shown as the mean with the standard deviation (*n* = 3, biological replicates).

## Data Availability

Not applicable.

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
