# Peer review of "Characterization of Light-Enhanced Respiration in Cyanobacteria"

_ijms, 2020, doi:10.3390/ijms22010342_

Round 1

Reviewer 1 Report

General comments:

This manuscript presents the light-enhanced dark respiration (LEDR) in cyanobacteria. Although LEDR has been observed in eukaryotic algae, its presence in cyanobacteria remains unknown. The authors report here that Synechocystissp. PCC 6803 exhibits LEDR, which is comparable to the capacity of the net photosynthetic oxygen evolution and is driven by the substrates produced during photosynthetic CO2 assimilation. The manuscript is well written and the interpretation of the results is consistent. The discussion provides deep and wide insight from the possible mechanisms of LEDR to its physiological roles and evolutional implications. In particular, I like the idea that LEDR compensates the high photosynthetic productivity that exceeds the demand for cell growth, via consumption of excess electrons.

Specific comments:

  1. The authors mention that the actual LEDR occurs during illumination, although LEDR can only be observed in darkness immediately after light is turned off, due to technical reasons. Actually, LEDR is disappearing soon in darkness. Then I am wondering if LEDR is no longer dark respiration but “light-enhanced respiration”. Please reconsider the naming of this phenomenon.
  2. 1: It looks like that respiration-derived electrons are transported to PSII. Modify this figure to match the context.
  3. Lines 74-75: Does ARTO reduce Cyd?
  4. Line 115: Insert “with” after “compete”, as this verb is not a transitive verb.
  5. 3 and 7B: Open symbols and lines are too faint. Avoid using gray colors.
  6. Line 197: Insert “further” before “increase”.
  7. Legend to Fig. 7: Replace “diamonds” with “triangles.
  8. Lines 217 and 218: Rewrite this sentence as it is vague. As in, “while in eukaryotic algae, it is mainly derived from LEDR where enough CO2 is available”.
  9. Line 221: “interdepend” should be replaced with other words, such as “depend mutually”.
  10. Line 227: It appears that the sentence “the large capacity of LEDR is purely originated from respiratory electron chain” might be contradictory to the main message that LEDR is motivated by the substrates produced during photosynthetic CO2 assimilation. More careful explanation is required here.
  11. Line 282: Does “increased light and CO2 conditions” mean high-light and high-CO2 conditions?

Author Response

Response to Reviewer 1

We thank you for reviewing our manuscript and providing helpful suggestions. We have revised our manuscript following your advice.

-----------------------------------------------------------------------------------------------------------------

  1. The authors mention that the actual LEDR occurs during illumination, although LEDR can only be observed in darkness immediately after light is turned off, due to technical reasons. Actually, LEDR is disappearing soon in darkness. Then I am wondering if LEDR is no longer dark respiration but “light-enhanced respiration”. Please reconsider the naming of this phenomenon.

-----------------------------------------------------------------------------------------------------------------

>>> We agree to the suggestion. In the revised manuscript, the term “LEDR” was changed to “LER” (line 46−47).

-----------------------------------------------------------------------------------------------------------------

  1. 1: It looks like that respiration-derived electrons are transported to PSII. Modify this figure to match the context.

-----------------------------------------------------------------------------------------------------------------

>>> We revised Figure 1.

-----------------------------------------------------------------------------------------------------------------

  1. Lines 74-75: Does ARTO reduce Cyd?

-----------------------------------------------------------------------------------------------------------------

>>> We revised the sentence (line 74−75).

-----------------------------------------------------------------------------------------------------------------

  1. Line 115: Insert “with” after “compete”, as this verb is not a transitive verb.

-----------------------------------------------------------------------------------------------------------------

>>> We inserted “with” (line 116).

-----------------------------------------------------------------------------------------------------------------

  1. 3 and 7B: Open symbols and lines are too faint. Avoid using gray colors.

-----------------------------------------------------------------------------------------------------------------

>>> We used closed symbols and different colors for these figures.

-----------------------------------------------------------------------------------------------------------------

  1. Line 197: Insert “further” before “increase”.

-----------------------------------------------------------------------------------------------------------------

>>> We inserted “further” (line 197).

-----------------------------------------------------------------------------------------------------------------

  1. Legend to Fig. 7: Replace “diamonds” with “triangles.

-----------------------------------------------------------------------------------------------------------------

>>> We corrected the legend.

-----------------------------------------------------------------------------------------------------------------

  1. Lines 217 and 218: Rewrite this sentence as it is vague. As in, “while in eukaryotic algae, it is mainly derived from LEDR where enough CO2 is available”.

-----------------------------------------------------------------------------------------------------------------

>>> We revised the sentence in reference to the suggestion (line 241−243).

-----------------------------------------------------------------------------------------------------------------

  1. Line 221: “interdepend” should be replaced with other words, such as “depend mutually”.

-----------------------------------------------------------------------------------------------------------------

>>> We changed the term (line 246).

-----------------------------------------------------------------------------------------------------------------

  1. Line 227: It appears that the sentence “the large capacity of LEDR is purely originated from respiratory electron chain” might be contradictory to the main message that LEDR is motivated by the substrates produced during photosynthetic CO2 assimilation. More careful explanation is required here.

-----------------------------------------------------------------------------------------------------------------

>>> We changed the sentence more concretely (line 251−254).

-----------------------------------------------------------------------------------------------------------------

  1. Line 282: Does “increased light and CO2 conditions” mean high-light and high-CO2 conditions?

-----------------------------------------------------------------------------------------------------------------

>>> We changed “increased” to “higher” (line 305).

-----------------------------------------------------------------------------------------------------------------

Reviewer 2 Report

The authors examined the light-enhanced oxygen consumption in eukaryotic algae in their past publications. Here the authors examined the similar physiological response in cyanobacteria. Experiments was carefully carried out, but the interpretation of the results seems to be somewhat biased. By comparing Figure 2 with Figure 7, the authors concluded that the enhancement effect cannot be solely ascribed to the accumulation of respiratory substrates produced by photosynthesis. To the eyes of the reviewer, however, most of the enhancement effect seems to be vanished by the addition of glucose. Interaction of respiration between photosynthesis in cyanobacteria has been established almost 40 years ago, and there is no sense of wonder in the observed phenomena. On the other hand, if the enhancement of oxygen consumption is regulated by step(s) other than the respiratory substrate as postulated in the discussion section, it is certainly interesting and worth to publish. For such discussion, however, most of the experiments should be carried out in the presence of glucose to remove the effect of the changes in respiratory substrates (just in the case of Figure 7).

As for introduction section, the description seems to be based on rather limited references. For example, the authors believed that TCA cycle is the main production points of reducing power (i.e. NAD(P)H), but there are several pieces of evidence that the oxidative pentose phosphate pathway is the main source of reducing power in cyanobacteria. Even though it is established that TCA cycle is not incomplete against the old belief, some metabolomic analysis indicates that flow through TCA cycle is negligible. The reviewer does not insist these are the only truth, but the authors' explanation seems to be rather one-sided. Furthermore, many of the historical papers that established the presence of interaction between photosynthesis and respiration in cyanobacteria are also not cited. Please look for more related papers with wide scope.

Author Response

Response to Reviewer 2

We thank you for reviewing our manuscript and providing helpful suggestions. We have revised our manuscript following your advice.

-----------------------------------------------------------------------------------------------------------------

By comparing Figure 2 with Figure 7, the authors concluded that the enhancement effect cannot be solely ascribed to the accumulation of respiratory substrates produced by photosynthesis. To the eyes of the reviewer, however, most of the enhancement effect seems to be vanished by the addition of glucose. Interaction of respiration between photosynthesis in cyanobacteria has been established almost 40 years ago, and there is no sense of wonder in the observed phenomena. On the other hand, if the enhancement of oxygen consumption is regulated by step(s) other than the respiratory substrate as postulated in the discussion section, it is certainly interesting and worth to publish. For such discussion, however, most of the experiments should be carried out in the presence of glucose to remove the effect of the changes in respiratory substrates (just in the case of Figure 7).

-----------------------------------------------------------------------------------------------------------------

>>> We agree to the suggestion that most of the enhancement effect seems to be vanished by the addition of glucose. However, what we would like to emphasize in this study is that the enhanced part during illumination was similar between in the absence and presence of glucose. In this revision, we attached a modified Fig. 7 for reviewing. In this figure, we normalized the O2 uptake rate at the point before the illumination. Respiratory O2 uptake was enhanced dependent on the illumination even in the presence of glucose.

              Additionally, we added the experimental results with glucose. The effect of exogenous glucose was also investigated at various light intensities (Supplemental Fig. S3), which is related to the experiment in Fig. 4 and also suggested that there is a certain impact of light on increasing the LER capacity regardless of the amount of glucose (line 204−208). Fig. 7 corresponds to the experiment with glucose related to Fig. 2 and 3. Both Fig. 5 and 6 show the mutant studies and we do not think that it makes senses to utilize glucose in these experiments. In the revised manuscript, we have additionally shown Fig. 8 for the experiment with glucose. In this experiment, we assessed the glucose effect on O2 uptake rate at different times after the actinic light was turned off (Fig. 8), implying a light-dependent activation factor for LER independent of exogenous glucose (line 216−224).

Based on these facts, in the revised manuscript we toned down but continued to propose that LER is regulated not only by respiratory substrates but also by illumination although the latter molecular mechanism is still unclear (e.g., line 279−284).

-----------------------------------------------------------------------------------------------------------------

As for introduction section, the description seems to be based on rather limited references. For example, the authors believed that TCA cycle is the main production points of reducing power (i.e. NAD(P)H), but there are several pieces of evidence that the oxidative pentose phosphate pathway is the main source of reducing power in cyanobacteria. Even though it is established that TCA cycle is not incomplete against the old belief, some metabolomic analysis indicates that flow through TCA cycle is negligible. The reviewer does not insist these are the only truth, but the authors' explanation seems to be rather one-sided. Furthermore, many of the historical papers that established the presence of interaction between photosynthesis and respiration in cyanobacteria are also not cited. Please look for more related papers with wide scope.

-----------------------------------------------------------------------------------------------------------------

>>> We added citations more based on a broad view (see the reference list in the revised manuscript). Then, we mentioned that oxidative pentose phosphate pathway produces reducing power in the Introduction section (line 62−65) and also the legend of Fig. 1.

----------------------------------------------------------------------

Round 2

Reviewer 2 Report

In this revised manuscript, the results in the presence of glucose were strengthened, making the authors' interpretation of the results more plausible. Outline of the work seems to be satisfactory. As for the interpretation of the results, the reviewer respects the opinion of the authors. Personally, however, the reviewer assumes that the regulation in LER in the presence of glucose is also metabolic one, since similar trends were observe in the effect of illumination time on LER with and without glucose (Figure 7B) as well as in the light saturation curve of LER with and without glucose (Figure S2). If possible, please discuss on this matter from such aspects.

Now that major problems are solved, please pay attention to the presentation of the figures for better understanding of readers. Some of the examples are shown below.

Figure 1: NADPH production was added to the legend, but not to the figure itself. Please also modify the figure.

Figure 2: The vertical axis represents oxygen evolution rate, not oxygen concentration. In that case, please put horizontal line at zero level (no oxygen production and consumption) to make the small but significant rate of respiration clear. The unit of the vertical axis is more appropriate to be "umol mgChl-1 h-1" istead of "umol mg-1 Chl h-1". These should be applied to the other figures.

Figure 3A: Please start vertical axis from zero level to make more intuitive comparison possible. This should be applied to the other figures.

Figure S3: Please use systematic use of symbols for better understanding for readers. You cannot intuitively understand the meaning of the symbols with black circles, blue triangles, purple reverse triangles and red diamonds. Instead, please use, for example, blue for the samples with glucose and red for the samples without glucose, and circles for net O2 evolution rate and triangles for LER. Preferably, such systematic use of symbols should be shared by all the figures. The legend to the horizontal axis should be "Photon flux densities" instead of "Light intensity". "photons" in the unit may not be necessary.

Author Response

We thank you for reviewing our manuscript and providing helpful suggestions. We have revised our manuscript following your advice.

-----------------------------------------------------------------------------------------------------------------

Personally, however, the reviewer assumes that the regulation in LER in the presence of glucose is also metabolic one, since similar trends were observed in the effect of illumination time on LER with and without glucose (Figure 7B) as well as in the light saturation curve of LER with and without glucose (Figure S2). If possible, please discuss on this matter from such aspects.

-----------------------------------------------------------------------------------------------------------------

>>> We agree that we cannot exclude the possibility that LER is metabolically regulated because LER might be stimulated only by a specific metabolite produced by photosynthesis but not by the glucose-induced metabolism. We note that Synechocystis 6803 increases the amounts of the glycolysis and Calvin-Benson cycle intermediates in dark in the presence of exogenous glucose (Shimakawa et al. 2014 Biosci. Biotechnol. Biochem. 78, 1997-2007). However, we cannot be sure that exogenous glucose metabolically mimics the situation where photosynthetic CO2 assimilation proceeds in light. This point was briefly mentioned at the final sentence of the 3.2 paragraph in Discussion section. In the revised manuscript, we further added the explanation for this (line 284−287).

-----------------------------------------------------------------------------------------------------------------

Figure 1: NADPH production was added to the legend, but not to the figure itself. Please also modify the figure.

Figure 2: The vertical axis represents oxygen evolution rate, not oxygen concentration. In that case, please put horizontal line at zero level (no oxygen production and consumption) to make the small but significant rate of respiration clear. The unit of the vertical axis is more appropriate to be "umol mgChl-1 h-1" istead of "umol mg-1 Chl h-1". These should be applied to the other figures.

Figure 3A: Please start vertical axis from zero level to make more intuitive comparison possible. This should be applied to the other figures.

Figure S3: Please use systematic use of symbols for better understanding for readers. You cannot intuitively understand the meaning of the symbols with black circles, blue triangles, purple reverse triangles and red diamonds. Instead, please use, for example, blue for the samples with glucose and red for the samples without glucose, and circles for net O2 evolution rate and triangles for LER. Preferably, such systematic use of symbols should be shared by all the figures. The legend to the horizontal axis should be "Photon flux densities" instead of "Light intensity". "photons" in the unit may not be necessary.

-----------------------------------------------------------------------------------------------------------------

>>> First, we revised Fig. 1 in reference to your advice. Second, we added horizontal lines at zero to y-axis in Fig. 2, 5B, 7A, and 8A. Third, we corrected "µmol mg-1 Chl h-1" to "µmol mg Chl-1 h-1" in all corresponding figures. Fourth, we tried to arrange the axis values and symbol styles as possible in all corresponding figures. We note that the revised Supplemental Fig. S3 is still complicated but we think that it is better to use four different colors considering the connection to Fig. 3A. Finally, we revised “light intensity” to “photon flux density” in Supplemental Fig. S3.

-----------------------------------------------------------------------------------------------------------------